# Effectiveness of a school-based programme of animal-assisted humane education in Hong Kong for the promotion of social and emotional learning: A quasi-experimental pilot study

**Joe T. K. Ngai** [ID]◉, **Rose W. M. Yu**◉, **Kathy K. Y. Chau**◉, **Paul W. C. Wong**\*◉

Department of Social Work and Social Administration, Faulty of Social Science, The University of Hong Kong, Hong Kong, Hong Kong

◉ These authors contributed equally to this work.
\* paulw@hku.hk

**Data Availability Statement:** All relevant data are within the paper and its Supporting Information files.

## Abstract

### Background

Humane education, which focuses on the cultivation of kindness and empathy towards animals, the environment, and fellow humans, helps children to be less egocentric and more sensitive to the human–animal interaction in ecology.

### Aim

This study aimed to evaluate an animal-assisted, school-based humane education programme that promotes a humane attitude and enhances social–emotional competence for children in Hong Kong.

### Method

A sequential mixed-methods formative evaluation was adopted in the pilot year of the programme. A controlled trial and focus groups were conducted to evaluate the preliminary outcomes and process of the programme and to identify the implementation obstacles and effective strategies. One hundred and ten primary three students from two primary schools participated in the study (55 in the intervention group and 55 in the control group with ordinary formal school extra-curricular activities). Paired sample t tests and a mixed ANOVA were conducted to explore the changes in students' social–emotional competence in our programme and two typical extra-curricular school programmes. Thematic analysis was conducted to categorise the transcriptions from the focus groups.

### Results

Quantitative findings indicated that class-based, animal-assisted humane education increased cognitive competence (t[24] = 2.42, p = .02), empathy (t[24] = 2.94, p < .01), and

**Funding:** The Keswick Foundation (http://www.keswickfoundation.org.hk) supported the development and implementation of "CARing Kids" programme. The author PWWC received the fund on behave of the research team. The funder had no role in study design, data collection and analysis, decision to publish, or preparation of the manuscript.

**Competing interests:** The authors have declared that no competing interests exist.

reduced hyperactivity (t[23] = -2.40, p = .02). Further analysis indicated that the participant recruitment strategies moderate the impact of interventions on the development of empathy (F[2,104] = 4.11, p = .02) and cognitive competence (F[2,104] = 2.96, p = .05). Qualitative analysis suggested three major themes: enhancement of self-control, promotion of humane attitude, and improvement of reading skills.

## Conclusion

The preliminary results of this pilot study indicate positive effects of the programme. Vigorous systematic formative evaluation on the process and effective implementation should be included in future follow-up studies to ensure its sustainability and fidelity.

## Introduction

Humans dominated the biosphere through urbanisation, industrialization, and globalisation in the last century [1]. Deterioration of our ecology evidenced by the degradation of fresh water, air pollution, and global warming frequently threatens the well-being of all living species [2]. In the past five decades, the number of people who live in urban areas has increased from 1.02 to 4.13 billion [3]. The rapid rate of urbanisation could create economic growth, improve education level, and enhance technology; however, it is also related to health hazards such as environmental pollution, infectious diseases, zoonoses and mental health issues [4, 5]. With the hope of continued existence of humanity in an environmentally sustainable society, the 'One Health' concept was proposed to advocate that future development of our world must focus on the complex interconnectedness and interdependence of all living species in a shared environment [6].

The One Health concept is a complex ideology incorporating ecology, evolution, environmentalism, medicine, and social sciences that translates into approaches for innovative and effective control of both infectious and multifactorial non-communicable diseases. Actualising this complex concept has many challenges [7]. In metropolises, we believe one of the challenges to operationalise the One Health concept is that many urbanised individuals have infrequent contact with animals, which may lead to a lack of understanding, sympathy, and appreciation for non-human animals and the ecosystem. Moreover, some individuals may have developed such strong senses of egocentrism and anthropocentrism that they are no longer sympathetic or compassionate towards fellow humans and non-human species. Research suggests that one's empathy and emotional affinity are significant factors affecting pro-environmental behaviours [8]. Hence, a possible initiative to actualise the One Health concept is to enhance people's empathy and care for others, including non-humans and the environment.

A substantial body of research has demonstrated that young children tend to have anthropocentric attitudes and reasonings more often than their older counterparts [9]. They show an egocentric tendency at their early cognitive and moral development stage [10, 11]. Therefore, education on empathy and prosocial behaviours through social and emotional learning (SEL) and humane education at a young age may help individuals to develop pro-environmental perspectives in adulthood; this is an essential initiative under the One Health concept.

A meta-analysis study of 213 SEL programmes found that SEL could significantly improve social behaviours, promote positive interpersonal and intrapersonal attitudes, lower distress level, reduce conduct problems, and enhance academic performance [12]. Moreover, studies also indicate that SEL programmes for children could have long-lasting impacts on the

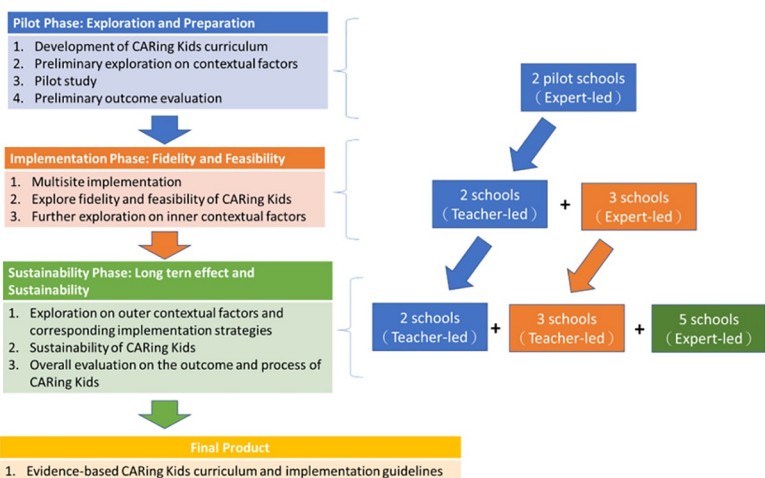

**Fig 1. Phases of the CARing Kids programme.**

educational and economic attainment, mental health, and sexual health of the participants when they become adolescents [13, 14]. In comparison, research on the effects of humane education is limited. Several studies suggest that humane education has a promising impact on the development of prosocial behaviour and empathy in early primary school students, and the effect can last for six to 18 months [15]. Criticisms of the evidence base of SEL and humane education are that the studies are too outcome-oriented, and the descriptions of the implementation processes are limited, and these factors threaten the recognition and generalisation of the findings [16, 17].

This paper describes the findings of a mixed-methods evaluation research study on the Competence in Active Resilience for Kids (CARing Kids) humane education with animal-assisted SEL implemented in primary schools in Hong Kong during its pilot year. The pilot year findings will inform the implementation and the sustainability phases of this three-phase programme. The pilot phase (Fig 1) aimed to develop a theoretically sound and feasible education programme based on existing literature and expert experiences and to estimate the feasibility and challenges for a larger-scale implementation phase by exploring contextual factors such as education and animal welfare situations in Hong Kong, as well as tapping into resources for programme sustainability. The second and third phases will focus on the implementation process and how to conduct the evaluation study on a larger scale. To enhance the sustainability of the CARing Kids, classes were taught by the research team in conjunction with the school teachers in the first year and then independently by school teachers in the second and third years. We reasoned that if teachers could be as effective as mental-health professionals in implementing programmes of this sort, there would be a significantly better chance for such programmes to be sustained at the participating schools, especially when funding was not available.

## Development of the animal-assisted SEL and humane education curriculum

The six-session CARing Kids programme was developed by a multidisciplinary team comprising academia, psychologists, and social workers; two team members received training from Pet Partners, and one is an animal-assisted therapist. Most SEL programme developers suggest that an instructionally sound and developmentally focused SEL curriculum should include

elements of emotional education, cognitive restructuring, interpersonal problem-solving, social skills training, empathy training, problem-solving, stress reduction and relaxation, and behavioural change [18]. There is no sound conclusion on the dosage effect of SEL. Some studies indicate shorter programmes have significant effects; however, the implementation quality appears to be a more crucial factor on SEL outcomes [18, 19]. Thus, it is believed that combining brief doses of each of the core elements in an instructionally sound manner will lead to improved SEL outcomes.

In our programme, the SEL framework was chosen as the theoretical foundation of the CARing Kids (Fig 2). According to SEL, a child's development takes place within a hierarchical and dynamic set of contexts [20]. Systematic influence and environmental factors such as school environment could have an impact on the outcome of both individual and programme outcomes. The CARing Kids programme not only comprises traditional SEL teaching on environmental literacy, social–emotional competence, cognitive competence, empathy, and interpersonal problem-solving but also includes a canine-assisted reading component with self-developed picture books for each session of the curriculum.

## Comparison of SEL, humane education, and CARing Kids

Although SEL, humane education and the CARing Kids programme share similar components, there are fundamental differences in terms of their learning objectives, programme emphasis, and implementation settings. In terms of learning objective, SEL has a broad spectrum of intended learning outcomes ranging from self-awareness to community involvement, whereas humane education particularly focuses on humane attitude, defined as 'kindness and compassion to environment, animals and fellows' [21]. The CARing Kids curriculum covers the fundamental component in SEL, but emphasizes empathy because we consider it is a building block of advanced social skills and humane attitude. In addition, we promote participants' awareness of environmental issues through true dog stories that happened in Hong Kong.

In terms of implementation in Hong Kong, SEL is often taught by professionals such as teachers, social workers, and school counsellors. However, humane education programmes in Hong Kong are generally not funded by the government and are mostly delivered by workers at animal shelters or animal welfare agencies as auxiliary work. These differences between settings impacted the design of the curriculum. For example, SEL can be delivered at schools as a semester-long curriculum while humane education is often a one-time seminar. Since the duration of humane education is generally short and mostly without any opportunity to interact with animals on site, participants might obtain hard facts on animal rights and welfare

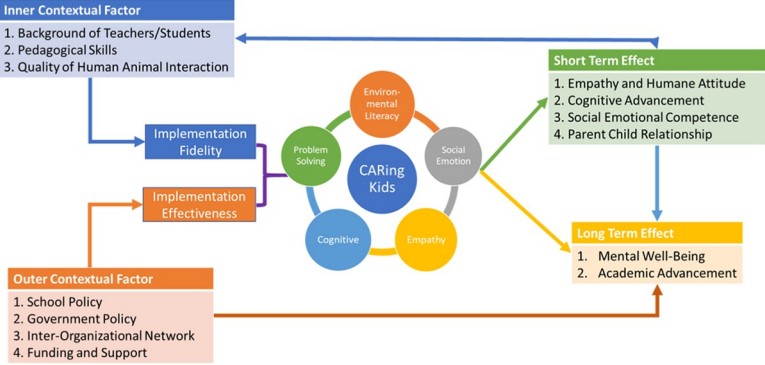

**Fig 2. The CARing Kids conceptual framework.**

without the chance to gain insight through experiential learning and personal reflections. The CARing Kids programme incorporates an animal assisted education component in six sessions that aim to provide experiential learning on humane attitude, SEL competencies and establishing a more meaningful bond with the reading canines. Further, the CARing Kids' curriculum design builds on the crucial learning elements of both SEL and humane education while ensuing the feasibility of implementation in school settings in Hong Kong. It is different from traditional SEL and humane education in terms of the spectrum of intended learning outcomes, implementation setting, and delivery methods.

## The picture books and the reading to the canine components

Picture books are commonly used in child education as they can help children visualise the stories. 'Stories' are compelling in humane education because they are compatible with addressing moral issues, provide context, create visual imagery, induce empathy, and inspire moral action [22]. Further, a picture book is a practical tool for teaching environmental concepts to young children and is effective in inducing behaviour change [23]. Furthermore, an international study revealed that Hong Kong students had the lowest reading motivation among 50 reviewed countries [24], therefore, we invited professional scriptwriters and artists to write six story books that were not only based on the themes of the curriculum (S1 Table) but also included the true stories of the canines who were also involved in the classroom teaching. These picture books are the backbone of the curriculum and enhance the reading motivation of the participating students. Reading to canines enables students to build up empathy, kindness, and a pro-animal attitude through direct interaction with the animals. All the canines in the CARing Kids are accredited by one of the four collaborating organisations experienced in providing animal-assisted interventions in Hong Kong. Only accredited canines passing the examination of their temperaments and reaction to other canines and children are considered suitable for working in a classroom setting [25].

## Overall structure of the CARing Kids programme

The CARing Kids curriculum is designed for grades 2 and 3 students to support the development of their SEL competence. Topics include environmental literacy, social emotional competence, cognitive competence, empathy, and interpersonal problem-solving. Each session includes five major parts: introduction, learning games, reading to canines, story discussion and conclusion. In each session, students read a picture book in the company of a reading canine followed discussion of the story. Each picture book carries a theme that corresponds to the theme of the session. Canine-companion reading is a core activity of the CARing Kids programme because we want to promote reading interest among Hong Kong students and provide opportunities for children to interact with a reading canine. Canine-companion reading helps to achieve the primary objective of the CARing Kids and enables participants to build up empathy and kindness through direct interaction with service canines. The topics and session objectives are listed in Table 1. A CARing Kids sample session is attached in SS2. As humane education aims to cultivate kindness and a pro-animal attitude, awareness, and considerations for animal welfare during the intervention should be taken seriously. The qualification of the handler and canine is a fundamental measure to ensure the safety of participants and an effective implementation process. To protect their welfare, the CARing Kids adheres to the code of ethics for service animals and the guidelines of the animal-assisted intervention [26, 27]. To the best of our knowledge, the current study is the first of its kind to evaluate animal-assisted education SEL with humane education in the region. It could therefore extend our existing

**Table 1. Comparison of implementation characteristics of the CARing Kids in two schools.**

|  | School 1 | School 2 |
|---|---|---|
| Participant recruitment | Random (class) | Teacher assigned (individual) |
| Grade | 3 | 3 |
| Intervention group | CARing Kids (class recruited) | CARing Kids (individually recruited) |
| Control group (TAU) | Life education | Elite talent training |
| Number of reading canines in each session | 3 | 3 |
| Time of each session | 75 minutes | 75 minutes |
| Number of participants in CARing Kids | 25 | 30 |
| Number of participants in control group | 25 | 30 |

knowledge on the sustainability and effectiveness of school-based, animal-assisted education programmes in non-Western cultures in the future.

## Methods

### Recruitment of schools

The CARing Kids programme obtained ethical approval from the Committee on the Use of Live Animals in Teaching and Research of the University of Hong Kong (CULATR 4838–18) and the Human Research Ethics Committee of the University of Hong Kong (EA1808020). The principals of the two participating schools were approached through the principal investigator's social network because these two schools were willing to trial innovative teaching ideologies or pedagogies to enhance students' learning and well-being. The research team approached the two principals and site visited the schools in July 2018, three months prior to programme implementation. The site visits aimed to understand the background of the school, characteristics of students, the socio-economic standard of the nearby neighbourhood, and each school's physical facilities to customise the feasible implementation plan. The research team presented the purpose of this programme, implementation procedures, and evaluation plan to the principals. After discussion with the two principals, two assignment methods were required to recruit students in the intervention group and the control group differently due to the different teaching arrangements and extra-curricular policies of the two schools.

Therefore, we adopted two different participant recruitment processes for the pilot study. School 1 participants were recruited using a class as a unit (i.e., invitations were sent to all students in one class). The intervention class participated in the CARing Kids curriculum, while the control class attended the ordinary school life education programme.

School 2 students were recruited at an individual level. The teachers at School 2 assigned one extra-curricular activity (e.g., sports, acrobatics, choir, scouting, dancing) to some students based on their interests and talents. This is called the 'Elite Talent Training programme' and acted as an active control. The Elite Talent Training programme is a routine school programme developed and implemented by School 2 during the weekly extra-curriculum lesson. The teachers select the students to learn a particular skill throughout the academic year, and some of them would represent the school to participate in various school competitions. The elite talent programme aims to enhance the students' self-confidence by providing them opportunities to express their talents through participating in intra- and interschool competitions. The intervention group of 30 students was randomly selected from those who did not join the Elite Talent Training programme. Another 30 students in the Elite Talent Training programme were randomly selected and recruited as an active control and attended the usual extra-curricular activities led by the school teachers. For the CARing Kids intervention group,

the instructor is a registered counselling psychologist who specialises in child education and animal-assisted intervention. Table 1 summarises the differences in implementation strategies between the two pilot schools.

## Evaluation methods

Formative mixed-methods evaluation was adopted to evaluate the impacts of the CARing Kids using a case-controlled trial methodology. Participants and their parents were invited to complete the questionnaires at three time points: before the programme commenced ($T_1$), one week after the programme ($T_2$), and four weeks after the programme ($T_3$). The research team prepared the data collection instruction scripts for the participating schools to ensure data quality. The school teachers read the script to instruct the participants to complete the questionnaire, and they read each item of the questionnaire in class to ensure the participants understood the meaning of the items. In addition, a research staff member monitored and assisted the entire data collection process.

Several focus groups with students, parents, and teachers were conducted at the end of the programme to review the impacts and the implementation obstacles in the school and community contexts. Purposive sampling was used to identify information-rich participants from diverse backgrounds. Participants were invited based on their verbal comprehension ability and involvement in the CARing Kids programme to ensure the quality and richness of the information. The school teacher initially explained the purpose of the study to the participants and their parents via phone invitation. A consent form and the details of this study were then clearly explained by the two experienced research staff before the focus groups began.

**Quantitative evaluation.** To examine the potential theory of pathway changes from the implementation of the curriculum to the changes in outcomes, we developed a conceptual framework that illustrates how the inner and outer contextual factors led to short-term and long-term outcome changes through the thoughtful implementation and education strategies of several essential components of the curriculum (Fig 2). Hence, the measures in the quantitative evaluation that aimed to explore the impacts of the CARing Kids curriculum on the development of SEL outcomes include the following measurement scales: First, a short version of the index of empathy for children and adolescents (IECA) was adopted [28]. It was used to measure the level of empathy towards others. Second, subscales of prosocial behaviour from the self-rated child version of the strength and difficulty questionnaire (SDQ) were chosen to evaluate the SEL competence of participants. Third, four items from the need for cognition scale (NCS) were extracted to evaluate individual cognitive competence, defined as 'the tendency for an individual to engage in and enjoy thinking' [29]. In the current sample, the reliability of the scales as indicated by Cronbach's alpha was found to be acceptable (short version of IECA: 0.72; prosocial behaviour subscale of SDQ: 0.77; and items from NCS: 0.79).

The parent questionnaire aimed to capture the parents' observations of the potential changes in their children's social and emotional competence. Hence, parents were also invited to fill in the questionnaires. The questionnaires were delivered through the school: the student participants received a parent questionnaire after completion of the student questionnaire in $T_1$, $T_2$, and $T_3$. Students delivered the parent questionnaire to their parents and returned it to the class teacher upon completion. The research team collected the parent questionnaires from the class teacher within one week; delayed submissions were not processed in the data analysis. The parent questionnaire included the subscale of parental closeness from the parent–child relationship scale short form (PCR-SF) to evaluate the parent–child relationship [30]. The reliability of the PCR-SF was found to be acceptable in the current sample ($\alpha = .785$).

Furthermore, the SDQ parent-report version was adopted to examine emotional problems and hyperactivity of participants. The reliability was 0.73 and 0.77, respectively.

**Qualitative evaluation.** Student, teacher, and parent focus groups were conducted to explore the process of SEL development and outer context implementation obstacles of the CARing Kids. The focus groups aimed to triangulate the quantitative results and explore the outer contextual factors that hinder the implementation process. The interview guide was drafted based on the proposed model to explore the impacts and the potential factors that affect the outcomes of the CARing Kids. The guide was then refined by the multidisciplinary research team to ensure validity. The student groups focused on the learning experience, self-perceived impact of the change of SEL outcomes, and school atmosphere; the parent groups focused on parental observations of participants' behaviour and family dynamics; and the teacher groups focused on the reported changes in students' learning, social–emotional behaviour, and implementation obstacles of the CARing Kids. Two research staff experienced in interviewing conducted the focus groups at the participating schools in December 2018, four weeks after the intervention was completed.

## Data analysis

For quantitative data obtained in questionnaires, paired sample $t$ tests were conducted to explore the short-term and long-term impacts of each intervention. A mixed ANOVA was then conducted to further explore the impact of the intervention and recruitment strategies in the changes in students' social–emotional competence. Each participant was assigned an identifying code generated from his or her background information on the questionnaires. Participants' responses were matched with their parent questionnaire and added to the analysis. In the paired sample $t$ tests, the dataset was split into four, based on the intervention received by the participant: (a) class recruited for the CARing Kids, (b) life education, (c) individual recruited for the CARing Kids, and (d) elite talent training. The paired sample $t$ tests were conducted separately to explore the change of mean difference of outcome variables between $T_1$, $T_2$, and $T_3$. In the mixed ANOVA, the datasets were combined: time acts as the within-subject factor with three levels (i.e., $T_1$, $T_2$, and $T_3$), and both recruitment methods (i.e., individual recruitment and class recruitment) and intervention (i.e., the CARing Kids and control) act as the between-subject factors. This analysis further explores how the intervention and recruitment method impact the outcome variables at different time points.

The qualitative data obtained in the focus groups were analysed by thematic analysis. First, we identified the semantic codes from the transcripts through the iterative process until all the content was accurately coded. During this process, the coder discussed the coding with the research team and the interviewers to develop a deeper understanding of participants' perceived learning experience, refine the coding process, and identify the interpretive codes that describe the underlying meaning of the data. Second, we compared codes from different focus groups to find recurrent codes and patterns. Third, by identifying the recurrent semantic codes and interpretative codes, we generated a list of themes and subthemes. To be categorised as a theme, supporting codes had to appear in the focus group from at least two parties of both schools. Fourth, the themes and full transcripts of all focus groups were reviewed and discussed by the multidisciplinary research team and interviewer upon achieving agreement on the final listed themes.

## Results

The CARing Kids curriculum was implemented between October 2018 and January 2019. The pilot study involved 110 student participants from two primary schools. Participants in both

schools shared a similar socio-economic background. Around half of the participants were male (51.82%), and most students were non-companion animal owners (87.27%). Most participating parents were mothers (70.00%) and completed junior secondary school (45.46%). The monthly household income ranged from HKD 10,000 to 39,999 (63.63%). The return rate of student questionnaires in $T_1$, $T_2$, and $T_3$ was 100%, 99%, and 100%, respectively. For the parent questionnaires, the return rate in $T_1$, $T_2$, and $T_3$ was 100%, 99%, and 89%, respectively. No missing values were obtained on the returned questionnaires. The dropout rate in parent questionnaires in $T_3$ was not statistically significant different among the different interventions ($\chi2$ = .04, df = 3, p = .50). Pearson chi-square test results indicated no significant statistical difference in terms of participants' gender ($\chi2$ = 4.95, df = 3, p = .18), parent gender ($\chi2$ = .64, df = 3, p = .88), parent education level ($\chi2$ = 11.50, df = 12, p = .49), household income ($\chi2$ = 9.68, df = 12, p = .64), or companion-animal ownership ($\chi2$ = 1.66, df = 3, p = .65) among the different interventions.

## Quantitative results

Table 2 summarises the mean and standard deviation of the outcome variables. Table 3 outlines the results of the paired-sample t test to explore the short-term ($T_2$–$T_1$) and long-term ($T_3$–$T_1$) impact of each intervention individually by comparing the within-group mean differences. The analysis indicated that only class-recruited CARing Kids appeared to have a significantly positive long-term impact on the development of empathy (t[24] = 2.94, p < .01) and cognitive competence (t[24] = 2.42, p = .02) among the four interventions. In addition, class-recruited CARing Kids showed a significant short-term reduction of hyperactivity in participants (t[23] = 2.40, p = .02). It is noteworthy that the captioned positive impact did not appear in other interventions, even in the individual-recruited CARing Kids group. It is assumed that participant recruitment strategies could impact the efficacy of the intervention.

To explore the moderation effect on the recruitment strategies, a mixed ANOVA was conducted. Table 4 presents the results of the mixed ANOVA to compare the impacts of participant recruitment methods (i.e., class vs individual) and groups (CARing Kids vs control) on the outcomes at three times points ($T_1$, $T_2$, and $T_3$). First, no significant main effect of the intervention was obtained, which indicates that the overall pooled impact of the CARing Kids (i.e., individually recruited, and class recruited) was not significantly different from the impact of the intervention in the control group (life education and elite training). Second, no significant main effect of recruitment strategy was obtained, suggesting that the participant

**Table 2.  The mean scores of social–emotional competences of participants at three time points.**

| | School 1 (class recruitment) | | | | | | | | | | | | School 2 (individual recruitment) | | | | | | | | | | | |
|---|---|---|---|---|---|---|---|---|---|---|---|---|---|---|---|---|---|---|---|---|---|---|---|---|
| | Intervention (CARing Kids) | | | | | | Control (formal life education) | | | | | | Intervention (CARing Kids) | | | | | | Control (elite training) | | | | | |
| | $T_1$ | | $T_2$ | | $T_3$ | | $T_1$ | | $T_2$ | | $T_3$ | | $T_1$ | | $T_2$ | | $T_3$ | | $T_1$ | | $T_2$ | | $T_3$ | |
| | M | SD | M | SD | M | SD | M | SD | M | SD | M | SD | M | SD | M | SD | M | SD | M | SD | M | SD | M | SD |
| **Student** | | | | | | | | | | | | | | | | | | | | | | | | |
| Prosocial | 19.60 | 3.94 | 19.28 | 3.57 | 19.88 | 3.61 | 18.40 | 5.12 | 20.12 | 4.34 | 19.6 | 3.52 | 18.03 | 4.07 | 19.40 | 3.82 | 18.70 | 4.50 | 17.93 | 3.40 | 19.66 | 4.2 | 19.3 | 5.07 |
| Cognitive Competence | 20.56 | 6.87 | 21.84 | 4.68 | 23.04 | 3.7 | 21.48 | 5.28 | 20.76 | 5.85 | 21.2 | 5.51 | 22.63 | 3.41 | 20.90 | 4.97 | 22.07 | 4.82 | 21.2 | 4.21 | 21.9 | 5.05 | 22.23 | 4.86 |
| Empathy | 12.16 | 3.39 | 13.60 | 3.57 | 13.64 | 3.39 | 12.24 | 3.11 | 13.24 | 3.73 | 13.08 | 3.63 | 14.90 | 3.33 | 13.07 | 3.46 | 13.50 | 3.33 | 12.60 | 3.87 | 14.34 | 3.69 | 13.50 | 4.58 |
| **Parent** | | | | | | | | | | | | | | | | | | | | | | | | |
| Emotional Problem | 11.12 | 2.80 | 10.92 | 2.81 | 11.32 | 2.75 | 12.60 | 3.83 | 13.20 | 3.38 | 11.64 | 3.26 | 11.87 | 3.85 | 11.73 | 4.49 | 11.07 | 3.38 | 11.00 | 3.97 | 10.33 | 3.54 | 10.89 | 3.28 |
| Hyperactivity | 14.64 | 3.93 | 12.67 | 3.36 | 12.79 | 3.74 | 15.16 | 3.99 | 15.24 | 3.11 | 15.64 | 3.58 | 15.20 | 4.51 | 14.13 | 4.49 | 14.21 | 3.81 | 15.13 | 4.06 | 14.27 | 3.81 | 14.79 | 3.63 |
| Closeness | 27.28 | 4.61 | 27.92 | 3.87 | 28.79 | 3.91 | 27.44 | 3.88 | 27.16 | 3.12 | 27.18 | 3.20 | 26.9 | 4.03 | 27.63 | 3.91 | 27.38 | 4.07 | 28.43 | 3.23 | 27.30 | 4.15 | 27.11 | 3.53 |

**Table 3. Paired sample t tests for the short-term and long-term impacts among different interventions.**

| | School 1 | | | | | | | | | | | | School 2 | | | | | | | | | | | |
| --- | --- | --- | --- | --- | --- | --- | --- | --- | --- | --- | --- | --- | --- | --- | --- | --- | --- | --- | --- | --- | --- | --- | --- | --- |
| | CARing Kids (class recruited) | | | | | | Life education | | | | | | CARing Kids (individual recruited) | | | | | | Elite talent training | | | | | |
| | Short term | | | Long term | | | Short term | | | Long term | | | Short term | | | Long term | | | Short term | | | Long term | | |
| | $T_2 - T_1$ | | | $T_3 - T_1$ | | | $T_2 - T_1$ | | | $T_3 - T_1$ | | | $T_2 - T_1$ | | | $T_3 - T_1$ | | | $T_2 - T_1$ | | | $T_3 - T_1$ | | |
| | t | p | d | t | p | d | t | p | d | t | p | d | t | p | d | t | p | d | t | p | d | t | p | d |
| **Student** | | | | | | | | | | | | | | | | | | | | | | | | |
| Prosocial | -.41 | .69 | -0.08 | .32 | .75 | 0.06 | 1.55 | .14 | 0.31 | 1.27 | .22 | 0.25 | 1.61 | .12 | 0.29 | .71 | .48 | 0.13 | **2.07** | **.05***  | **0.38** | 1.53 | .14 | 0.28 |
| Cognitive Competence | 1.09 | .29 | 0.22 | **2.42** | **.02*** | **0.48** | -.50 | .62 | -0.10 | -.25 | .81 | -0.05 | -1.87 | .07 | -0.34 | -.75 | .46 | -0.14 | .26 | .80 | 0.05 | 1.38 | .18 | 0.25 |
| Empathy | **2.90** | **.01*** | **0.58** | **2.94** | **.01*** | **0.59** | 1.85 | .08 | 0.37 | .90 | .38 | 0.18 | -1.90 | .07 | -0.35 | -1.82 | .08 | -0.33 | 1.86 | .07 | 0.35 | 1.59 | .12 | 0.29 |
| **Parent** | | | | | | | | | | | | | | | | | | | | | | | | |
| Emotional problem | -.41 | .69 | -0.08 | .36 | .72 | 0.08 | -.80 | .43 | -0.16 | -1.44 | .16 | -0.31 | -1.08 | .29 | -0.20 | -.79 | .44 | -0.15 | -.22 | .83 | -0.04 | .25 | .80 | 0.05 |
| Hyperactivity | **-2.40** | **.02*** | **-0.49** | -1.01 | .33 | -0.23 | .14 | .89 | 0.03 | .73 | .47 | 0.16 | -1.80 | .08 | -0.33 | -1.17 | .25 | -0.22 | -1.55 | .13 | -0.28 | -.31 | .76 | -0.06 |
| Closeness | .67 | .51 | 0.14 | .99 | .33 | 0.23 | -.44 | .66 | -0.09 | -.19 | .85 | -0.04 | 1.27 | .21 | 0.23 | 1.20 | .24 | 0.22 | -1.52 | .14 | -0.28 | **-2.17** | **.04*** | **-0.41** |

* < .05.

recruitment method alone (class recruited vs individual recruited) did not have a significant impact on the development of SEL outcomes. Third, significant interaction effects were found in cognitive competence (F[2,104] = 2.96, p = .05) and empathy (F[2,104] = 4.11, p = .02) in the analysis. It is suggested that the intersectionality between participant recruitment method (i.e., class-based vs individual-based) and interventions could impact the empathy and cognitive competence of participants.

In a comparison of the different trends in empathy and cognitive development between individually recruited and class-recruited CARing Kids groups, a decreased level of empathy and cognitive competence in $T_2$ only appears in the individually recruited CARing Kids group. Nonetheless, the scores rebound at $T_3$ (Fig 3). Compared with all other interventions, individually recruited CARing Kids were the only group receiving the curriculum that required intensive collaboration for group learning activities (e.g., group reading with companion canines) with unfamiliar classmates. It is assumed that a temporary reduction of empathy and cognitive competence in $T_2$ indicates that the participants in this situation need extra time to develop

**Table 4. Results of mixed ANOVA of the Caring Kids and recruitment method on outcome variables at different time points.**

| | Times | | Times * intervention | | Times * recruitment method | | Times * recruitment method* intervention | |
| --- | --- | --- | --- | --- | --- | --- | --- | --- |
| | F | p | F | p | F | p | F | p |
| **Student** | | | | | | | | |
| Prosocial | **3.50** | **0.03*** | 0.90 | 0.41 | 0.35 | 0.70 | 0.79 | 0.46 |
| Cognitive competence | 1.74 | 0.18 | 0.20 | 0.79 | 0.56 | 0.55 | **2.96** | **0.05*** |
| Empathy | 1.58 | 0.21 | 2.29 | 0.10 | 1.66 | 0.19 | **4.11** | **0.02*** |
| **Parent** | | | | | | | | |
| Emotional problem | 0.40 | 0.67 | 0.10 | 0.91 | 0.30 | 0.74 | 2.68 | 0.07 |
| Hyperactivity | 2.65 | 0.73 | 1.00 | 0.37 | 0.34 | 0.71 | 0.48 | 0.62 |
| Closeness | 0.07 | 0.93 | 2.51 | 0.08 | 0.70 | 0.50 | 0.43 | 0.65 |

* < .05.

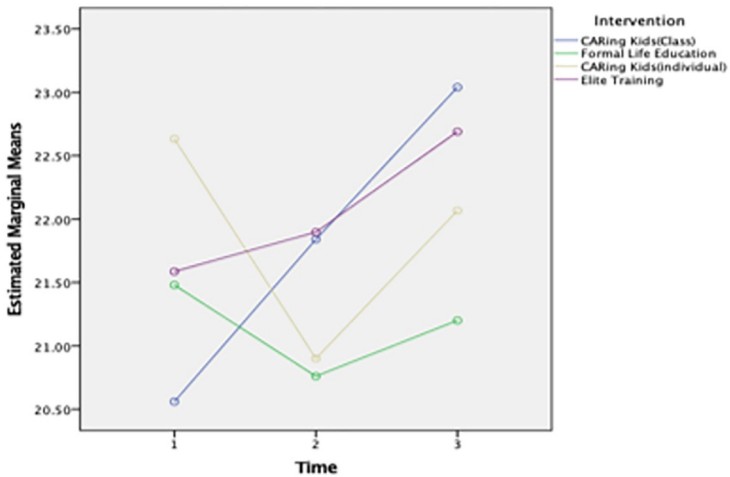

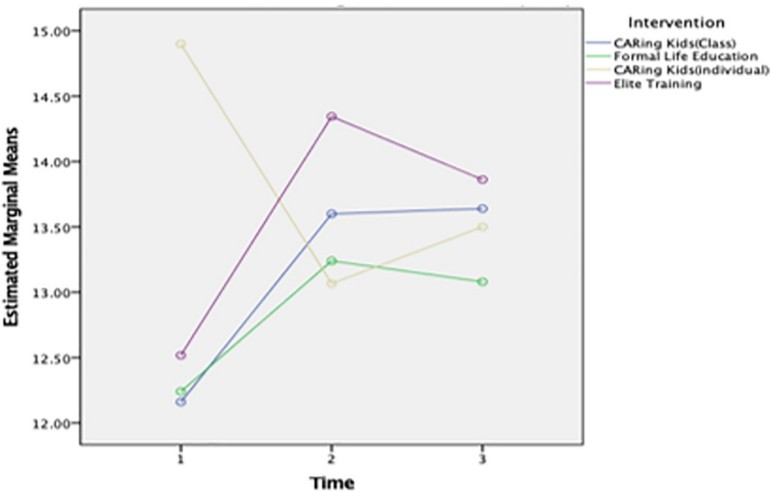

**Fig 3.** Comparison of the mean scores of socio-emotional competences among different interventions in different time points (a and b).

the social connectedness and collaborative learning dynamics with schoolmates. Therefore, the intervention effect was delayed and resulted in the rebound at $T_3$ in the individually recruited CARing Kids group. The results indicate that the recruitment strategies could alter the outcomes for the same intervention in practical implementation settings.

In this quasi-experimental setting, the moderation variable–participant recruitment strategies–could potentially overlap with school factors. With regard to the similar student background and regional characteristic of the two schools in this study, we assumed that the sources of the major moderation effect that impacted the outcomes were the different participant recruitment and programme implementation methods (i.e., class-based recruitment vs individual-based recruitment). The participants recruited individually took part in the CARing Kids with unfamiliar schoolmates, resulting in the delayed impact of SEL outcomes.

In summary, the findings of the paired-sample t test and mixed ANOVA indicate that school factors (recruitment strategies, in this case) play an important role in moderating the intervention effect on cognitive competence and empathy development. The evidence indicates that the CARing Kids works best for promoting SEL competence if implemented at a class-based level.

## Qualitative results

Eight students, seven teachers, and eight parents were interviewed in six semi-structured focus groups. All the interviews were audio-recorded, transcribed, and analysed using MAXQDA 2020. Three major themes of the effects of the CARing Kids were identified: enhancement of self-control, advancement of a humane attitude, and improvement of reading skills.

**Enhancement of self-control.** All parties reported enhancement of students' self-control in three areas: (a) increased self-disciplinary behaviour during the CARing Kids lessons; (b) reduced aggressive behaviour in school and home settings; and (c) improved anger management when faced with problems.

**(a) *During the CARing Kids lesson: Self-disciplinary behavior.*** Teachers from both schools reported that student participants appeared to be more 'manageable' in the CARing Kids lesson. Teachers stated that this phenomenon is related to their motivation to interact with the canines. The following dialogue outlines the teachers' observations in CARing Kids.

*Teacher A*: *I think this programme can benefit the students with conduct problems the most. It is because they want to interact with the dogs, they, therefore, are willing to comply with the rules.*

*Teacher B*: *Not only the students who were diagnosed with special education needs, but even the students who often have conduct issues [benefited]. They were motivated in this lesson, so they were more likely to follow the lesson rules.*

*Teacher C*: *Based on my observations on their [students'] daily behaviour, I am surprised that those students can follow the instructions quietly and effectively.*

*Teacher D*: *I noticed that some students become more 'peaceful' and able to follow [the rules]. Maybe they want to interact with the canines. I found this programme can be especially helpful to students with special education needs.*

This phenomenon was also reported in student and parent focus groups. The teachers stated that even though participants felt excited in the lesson, they were more capable of containing their excitement and behaved more appropriately in the CARing Kid lesson.

**(b) *Within school and family: Reduction of aggressive behaviour.*** Participants in CARing Kids in both schools reported a reduction of physical aggressiveness such as fighting and bullying in daily schooling. The following focus group members' dialogue describes the situation.

*Student A*: *I have some changes in a peer relationship. I used to hit people very often, and now I do not [when I argue with a classmate]. I will talk to my mother and will not hit them anymore.*

*Student B*: *I also notice that Student C used to bully others very often in the physical education lesson, but he stopped fighting with classmates now.*

**(c) *Beyond the classroom: Improvement in anger management.*** The parent focus group revealed that students acquired better anger management skills. It appeared that the noticeable

improvement related to the acquisition of new problem-solving skills and interpersonal conflict resolution induced by positive human–animal interaction. The excerpt below describes what one of the members of the parent focus group observed.

> *Parent A*: *My daughter is easily irritated. If she finds something she cannot handle, then she will be furious. I discovered that her emotions are more stable now. She is calmer than before. When she faces a problem, she stops and thinks about solutions. She told me that dogs and humans could interact peacefully. I found her most significant improvement is that she is less angry than before. For example, [before the programme] she was short-tempered. When she played with a doll, she would cut off the hair if the hairstyle did not fulfil her expectation.*

The participants' class teachers also reported improvement in emotional stability during the CARing Kids lessons. Even the most hot-tempered participants did not argue with peers during the lessons.

**Advancement of a humane attitude.** All the focus groups reported the students acquired a more humane attitude through the programme. Students developed a positive perception of canines, which was observed in their human–canine interactions. Teachers also stated that most students changed their negative stereotype towards canines. Moreover, several parents reported that their children would initiate the conversation related to the CARing Kids curriculum (i.e., responsible animal ownership, the contribution of a service animal) with family members and that the humane attitude was transferred to the family. Some parents pointed out that their children teach them how to use the techniques learned in the CARing Kids curriculum to overcome the fear of canines. For example:

> *Parent A*: *I would be petrified whenever I saw a dog because of intense fear. Now, my daughter teaches me what I can do to interact with a dog. I am pleased that she shared that information with me. I also feel she is less irritated than before.*

Occasionally, participants reported that their perception of the animal community changed after joining the CARing Kids programme. They often developed a sense of sympathy towards the animal community, as shown below.

> *Student*: *A black dog was living in my neighbourhood. It always argued with me [barked at me]. I think it was dangerous. Two months ago, I found out that my neighbour poisoned it. I used to hate it very much. However, I feel resentful now because it is a life, even if it is a dog.*

**Improvement of reading skills.** The focus groups revealed that canine companion reading could enhance participants' reading motivation and attitude. The teachers stated that participants had stronger reading motivation in the CARing Kids compared with other lessons. Interacting with canines appeared to be a powerful reinforcer of the development of reading habits. The results reveal that a small group reading setting appeared to be effective in promoting a positive reading attitude and social skills. The transcripts below illustrate teachers' observations.

> *Teacher A*: *I noticed that students were more involved when they read the story to the service dog. Even some students who did not like to speak or read were motivated when they could pet the service dog while reading.*

> *Teacher B*: *Although the parent said that their children had limited verbal comprehension ability, their reading motivation was significantly higher compared to the formal lessons.*

The parent focus group members indicated that participants' reading motivation is sustainable. Positive companion reading experience seems beneficial to the parent–child relationship.

*Parent A*: *My child has a problematic relationship with his father. However, he tried to read the CARing Kids storybooks to his father. I found their relationship has improved. This birthday I brought two storybooks to him, and he started to read the story with me.*

Empirical evidence obtained in the focus groups supports and extends the findings of the quantitative analysis, shedding light on the development of social–emotional competence and the factors that facilitate the process. The quality of positive human–animal interaction appears to be a crucial factor in this setting. As described in the focus groups, the participants' positive experience interacting with the service canines could provide insight into their interpersonal style, resulting in new coping strategies to handle interpersonal conflicts.

**Outer context implementation obstacles.** Members of the teacher focus group described several outer contextual implementation obstacles that threaten the efficacy and the feasibility of the CARing Kids. Referring to the conceptual model of the CARing Kids, teachers' descriptions were related to the education policy and community culture factors.

**(a) Education policy: Packed school schedule.** Both parents and teachers commented that the six sessions of the CARing Kids intervention appeared short and the participants were unable to attain substantial improvement. Some parents stated that they hoped this intervention could be conducted throughout a whole academic year. However, the teachers noted that the school might not have the capacity to conduct a year-long extra-curricular programme because of the packed school schedule.

**(b) Community culture: Insufficient number of trained canines.** Positive human–animal interaction experience appears to be essential for the development of social–emotional competence in the CARing Kids. However, the number of dog owners in Hong Kong is far below international counterparts, and this phenomenon would increase the implementation difficulties. The number of participants in each class was around 30. The human-to-canine ratio resulted in limited interaction time between participants and service animals.

Table 5 summarizes the overall results of the qualitative analysis; the Global themes, organized themes and the codes were presented.

**Table 5. Themes that emerged from qualitative analysis.**

| Global themes | Organised themes | Codes |
|---|---|---|
| Effect of the CARing Kids | Enhancement of self-control | Self-disciplinary behaviours |
| | | Reduced aggression |
| | | Anger management |
| | Advancement of humane attitude | Responsible companion animal ownership |
| | | Perception of community animal |
| | | Empathy of animal |
| | Improvement of reading skills | Reading confidence |
| | | Reading motivation |
| Outer context implementation obstacle | Education policy | School schedule |
| | | Diverse student needs |
| | Community culture | Dog-ownership |
| | | Neighbourhood animal friendliness |

## Discussion

It is understood that the main concern of the One Health concept is health [4], but this understanding should not be restricted to the biological or chemical components. How humans perceive themselves in the ecosystem depends mostly on their anthropocentric beliefs. Biophilia is a common human need and essential for maintaining mental well-being [31]. Children should be given a chance to interact with nature, especially in this highly urbanised metropolis. Hence, one of the aims of our programme is to provide an opportunity for primary students to be less egocentric. Children in Hong Kong have few chances to interact with an animal compared with their international counterparts. Furthermore, given that primary students in Hong Kong have a packed academic schedule and prolonged study hours [32], our programme allows the students to have a relaxing and stress-free learning environment to alter their negative perception of readings, cultivate empathy, and promote a humane attitude. Based on the preliminary findings of this pilot study and our school observations, the CARing Kids achieved these primary objectives. Moreover, this pilot experience provides insight into how the presence of animals impacts the process of development of empathy and social–emotional competence in the school setting in Hong Kong.

### Learning motivation in the CARing Kids

The current results echo previous literature on animal assisted education in the education setting, indicating that students tend to have a higher level of learning motivation with the presence of animals in the classrooms [33]. As revealed in the focus groups, the students' learning motivation was related to their strong tendency to interact with the canines. This tendency enhanced their learning capacity to overcome the daily educational obstacles and tolerate learning frustration. The affiliation with the canines in the CARing Kids programme can be attributed to two factors. First, the biophilia hypothesis, referring to the innate tendency to connect to all that is alive and vital [34], and second, the high level of curiosity and novelty caused by the insufficient exposure to live animals in the metropolis. The reading to a canine component was an entirely new learning experience that allowed the students to learn in a stress-free and non-judgmental environment. This setting might contribute to participant's reading motivation and enhance reading competence. In addition, human–animal interaction also changed participants' negative stereotypes of animals. Participants acquired a positive humane attitude–compassion for other animals–as revealed by their enhancement of empathy. Further, participants considered all animal life as equal and deserving of respect, as described in the student focus group.

### Calming effect, self-control, and classroom atmosphere

The data indicates that the CARing Kids could promote students' self-control and facilitate a positive learning atmosphere. People with high self-control experience more positive emotions, life satisfaction, and happiness [35]. As a long-term impact, self-control can help participants strive for personal goals and obtain a higher level of life satisfaction. Moreover, students' self-control ability can impact the learning environment. Students with a high level of self-control can resist impulsivity and maintain attention in learning situations. Therefore, teachers can, implement more effective classroom management strategies. One of the possible reasons for the improvement in self-control can be attributed to direct physical contact with the service animals. Petting an animal can induce relaxing physiological and neurological responses such as reduction in systolic blood pressure, lower heart rate, and reduction of cortisol, resulting in stabilising emotions and a calming effect [36–38]. In the pedagogical perspective, students with stable emotions made the classroom more manageable. The programme provides an

effective teaching and learning environment for both the teachers and students. Participants, therefore, learned the social–emotional skills in the curriculum in a relatively efficient context and were more likely to acquire the skills to cope with their frustration when facing daily challenges in both school and family settings.

## Implementation obstacles and strategies for a humane education programme

One of the main purposes of this pilot study was to identify the implementation obstacles and corresponding strategies for this new format of animal-assisted humane education in Hong Kong. The formative mixed-methods evaluation revealed several core contextual obstacles that would affect the effectiveness of the programme implementation. One of the findings revealed in the quantitative evaluation is that intersectionality between school and intervention could affect the process of empathy and cognitive development. For example, School 1, which adopted class-based implementation for the CARing Kids, tends to yield more positive results when compared to the individual-based recruitment in School 2.

The critical component to enhance the efficacy of the programme is the creation of a prosocial and empathetic atmosphere in classrooms. SEL encompasses the promotion of a positive classroom and the building of a safe and caring school environment that encourages students' learning. Hence, SEL concerns not only individuals' social and emotional development but also the social and emotional climate of classrooms and schools. Based on the proposed model, programme effectiveness would be affected by inner contextual factors such as the classroom and school characteristics.

The effect of the inner contextual factor could override the intrapersonal factor at a significant level in the education programme. For School 1, which recruited the participants at the class level, all the participants shared the same learning experience. Synchronicity in learning facilitates a positive learning atmosphere and helps to establish a prosocial classroom culture. As the classroom is an important social context for primary students, participants in School 1 could find it easier to apply the learned social–emotional skills to handle interpersonal issues. The reciprocal prosocial interaction among classmates reinforces participants' willingness to practice the acquired social skills and eventually attain better SEL outcomes.

Moreover, this study revealed the importance of contextual factors, such as school policy and community culture, on prosocial behaviour development. However, these factors are often ignored in an experimental research setting. The CARing Kids programme systematically increased the involvement of the frontline practitioners (i.e., teachers, animal-assisted service providers) in the research process and development of the teaching materials to ensure practicability. Effective implementation of animal-assisted education requires the appropriate protocols; careful planning between the handlers, principals, and school staff; and reflective practice [24]. A lack of collaboration of the above parties may result in ill-planned and ill-implemented practices that fail to establish positive human–animal bonding and may be harmful to students' and service animals' health and welfare. The neglect of a realistic contextual factor often fails to transfer research findings into practice. Collaboration among the multidisciplinary professionals is crucial to identify the critical implementation obstacles and generate strategies to overcome the difficulties in programme implementation in the school setting. The cooperation of the teachers, handlers, animal organisations, and researchers enhances the applicability of the CARing Kids programme, transforming results obtained in this pilot study into a feasible programme that can be adopted in a realistic school setting.

It is noteworthy that the CARing Kids holds both animal welfare and students' learning outcomes at the same level of importance. It is of concern that there is an increasing number of studies focusing on the impact of animal-assisted education, yet only a few mentioned the importance of animal welfare and the implementation strategies to preserve the animals' rights [39]. One of the principles of humane education is to cultivate kindness towards animals; hence, practitioners and researchers should not overlook the stress on animals during the intervention. Positive human–animal bonding is an essential process for empathy and prosocial development in humane education. Although the contextual factors such as education policy and community culture make the implementation of animal-assisted humane education challenging in Hong Kong primary school settings, the close cooperation of multidisciplinary professionals could compensate for the contextual disadvantages.

## Insight, limitation, and further research direction

Promoting evidence-based practice in the community is challenging and requires a long period of investment. For this pilot study, we attempted to accomplish this task in three years. Additional effort in risk assessment, practical support, and quality monitoring was made to facilitate the research and implementation process. Nonetheless, developing an evidence-based intervention protocol and promoting the evidence-based practice of animal-assisted humane education require additional effort. The three recommendations below will be addressed in the upcoming phase of the research.

**Evaluation of human-animal bonding.** This study provides a preliminary result from the CARing Kids programme. However, the data obtained to date focus on the SEL outcomes. In terms of human-animal bonding, the intermediate variables were not robustly monitored during this study because of the contextual constraints. Even though the participants revealed the potential impact of the human–animal bonding in the focus groups, the objective evaluation shall be included in future study phases to justify the impact of the human–animal bonding in developing SEL and humane attitude in the CARing Kids programme.

**Relationship and mechanism between human–animal bonding, social–emotional competence, and inner contextual factors.** Although this study revealed positive development of SEL competence among participants, the mechanism remains uncertain. How individual- and systematic-level factors such as school support, teaching styles, and community factors could interplay and impact the effectiveness and implementation efficacy of animal-assisted education will be explored in a future study. This will consist of an in-depth case study of the participants and will be conducted in the implementation phase to document the process of changes and the underlying mechanism.

**Effective implementation and programme sustainability.** Both schools in this pilot study were located in a district with similar socio-economic status and population composition. The background of the school could impact the effectiveness of implementation. Therefore, multi-sited intervention research will be conducted in the implementation phase. The impact of organisational characteristics, including the climate and leadership, which play an essential role in the effective implementation and sustainability of the CARing Kids, will be evaluated in the sustainability phase.

## Conclusion

This study obtained preliminary evidence on the impact of the newly developed animal assisted humane education programme. The findings indicate that our programme could be able to facilitate the development of SEL and humane attitude. We believe that humane attitude is crucial for the students to assimilate the concept of 'One Health' in later life. One of the

purposes of introducing the 'One Health' concept to students is to tackle the problems associated with the negative impacts of over-urbanisation. The rate of urbanisation in the last two decades has been rapid, and we are now witnessing human-manufactured change at an unprecedented rate in human history [40]. In addition to the explosive growth of economies and technological advancement, urbanisation is accompanied by pollution, destruction of the environment, and alienation from nature. Although urbanisation only covers around 5% of the total landmass, it consumes around 70% of global energy and impacts 95% of the remaining land globally [41]. Humans have a blind assumption that human beings have an unlimited capacity to adapt to the environment [31]. Nevertheless, studies and the recurrent outbreak of zoonoses seem to suggest that our adaptive capacity is limited and, in fact, far below what we have assumed.

Since the natural environment is as central to human history as social behaviour itself, we cannot underestimate the impact of 'nature alienation' on human well-being [34]. Marian Wright Edelman, a prominent activist for children's rights, defines education as: '*Education is for improving the lives of others and for leaving your community and world better than you found it*' [42]. As such, animal-assisted humane education is a possible approach to compensate for some of the costs of urbanisation and an essential step to promote the One Health concept. Nonetheless, the past 150 years of the history of humane education suggests that sound scientific evidence is the foundation for the development of any discipline of knowledge. Sound research evidence, evidence-based practice, and professional training of the programme implementor are necessary conditions for the active promotion of animal-assisted humane education. Compared with traditional humane education, animal-assisted humane education is in its infancy. Our understanding of how animal-assisted intervention impacts the welfare of service animals is still limited. Exploring how the service animal receives ethical treatment during the intervention is one of the focus areas of animal-assisted education research [43].

The development of the CARing Kids programme aims to compensate for the shortcomings of the current education system, including an over–emphasis on academic achievement and an underestimation of the importance of social–emotional competence and concern for environmental sustainability in Hong Kong. One Health is the solution to ensure sustainable environmental and human health under the inevitable global urbanisation trend. We believe that humane education in childhood is a foundation for developing the One Health mindset. However, traditional humane education is not a priority for primary schools in Hong Kong.

While it appears that implementing traditional humane education in primary schools is challenging, the CARing Kids uses an innovative approach to promote humane attitude and SEL outcomes. The CARing Kids has created a first-person human–animal interaction experience to cultivate a humane attitude in students. We also collect stories from the local community and create reading materials to ensure the students apply their social-emotional competencies in real life. This study's preliminary results indicate that it is feasible and that there are benefits in implementing this new form of humane education in a formal school setting.

Further, the CARing Kids programme is a research-driven programme to facilitate the evidence-based practice of animal-assisted humane education in a formal school setting. Series of formative evaluation research will be conducted to inform the programme implementor of the most effective, efficient, and sustainable way to deliver the CARing Kids curriculum. Additional interdisciplinary efforts are required when introducing and sustaining any new form of education programme in an existing education system. The inner and outer contextual factors (e.g., teachers' pedagogical skills, government policy, community culture) will impact the effectiveness of the implementation of this newly developed animal-assisted humane programme. Even though contextual disadvantages are inevitable, the cooperation of multidisciplinary

professionals (i.e., researchers, teachers, and handlers) can compensate for the contextual disadvantages. This pilot study's preliminary result indicates that the CARing Kids programme has a positive impact on enhancing participants' empathy, self-control, emotional regulation, and humane attitude. More importantly, this study provides strong evidence that the CARing Kids curriculum is as effective as the mainstream social–emotional intervention in primary schools if implemented with the appropriate strategy and animal friendly and humane educators and advocators.

## Supporting information

**S1 Table. Summary of the CARing Kids programme.**
(TIF)

**S1 File. Curriculum of the CARing Kids lesson 1.**
(PDF)

**S1 Data.**
(SAV)

## Author Contributions

**Conceptualization:** Joe T. K. Ngai, Rose W. M. Yu, Paul W. C. Wong.

**Data curation:** Joe T. K. Ngai, Rose W. M. Yu, Kathy K. Y. Chau.

**Formal analysis:** Joe T. K. Ngai.

**Funding acquisition:** Paul W. C. Wong.

**Methodology:** Joe T. K. Ngai, Paul W. C. Wong.

**Project administration:** Rose W. M. Yu, Kathy K. Y. Chau.

**Resources:** Paul W. C. Wong.

**Software:** Joe T. K. Ngai.

**Supervision:** Paul W. C. Wong.

**Validation:** Joe T. K. Ngai, Rose W. M. Yu.

**Visualization:** Joe T. K. Ngai.

**Writing – original draft:** Joe T. K. Ngai, Paul W. C. Wong.

**Writing – review & editing:** Rose W. M. Yu, Kathy K. Y. Chau, Paul W. C. Wong.

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
