## [Decision Letter · Decision Letter 0]

30 Nov 2020

PONE-D-20-25086

Effectiveness of a school-based programme for promotion of social and emotional learning of an animal-assisted humane education in Hong Kong: A quasi-experimental pilot study

PLOS ONE

Dear Dr. Ngai,

Thank you for submitting your manuscript to PLOS ONE. After careful consideration, we feel that it has merit but does not fully meet PLOS ONE’s publication criteria as it currently stands. Therefore, we invite you to submit a revised version of the manuscript that addresses the points raised during the review process.

I feel your manuscript is almost accepted, but you should introduce the improvements highlighted by Reviewer #1.

We look forward to receiving your revised manuscript.

Kind regards,

Juan-Carlos Pérez-González, Ph.D.

Academic Editor

PLOS ONE

Journal Requirements:

2. Please provide additional information about your teaching intervention, such as detailed curriculum, description of texts or methods used, or supporting educational material (such as the story books) that would allow others to replicate your study. If materials, methods, and protocols are well established, authors may cite articles where those protocols are described in detail, but the submission should include sufficient information to be understood independent of these references (https://journals.plos.org/plosone/s/submission-guidelines#loc-materials-and-methods). Please note that any materials submitted should not be under a copyright more restrictive than CC-BY.

Reviewers' comments:

Reviewer's Responses to Questions

**Comments to the Author**

1. Is the manuscript technically sound, and do the data support the conclusions?

Reviewer #1: Partly

Reviewer #2: Yes

2. Has the statistical analysis been performed appropriately and rigorously? 

Reviewer #1: Yes

Reviewer #2: Yes

3. Have the authors made all data underlying the findings in their manuscript fully available?

Reviewer #1: No

Reviewer #2: Yes

4. Is the manuscript presented in an intelligible fashion and written in standard English?

Reviewer #1: Yes

Reviewer #2: Yes

5. Review Comments to the Author

Reviewer #1: This manuscript presents very interesting hypotheses about socio emotional learning strategies and their applicability in school-based programs, linked to a thorough research methodology. Authors have made an excellent use of quantitative and qualitative analysis to verify not only the effectiveness of the program, but its feasibility and sustainability over time, including context-related factors in the process. Furthermore, language, structure and presentation of data show the quality expected in a rigorous scientific journal.

Nevertheless, some aspects should be reviewed before publication:

- SEL theoretical framework: As SEL has proven to be effective in abundant research, it has also been criticized by its excessive broadness and variation between studies, which hinders validity and replication (Pérez-González & Qualter, 2018; Zeidner, Roberts, & Matthews, 2002). Authors state to choose SEL as the “theoretical foundation” of the CARing Kids programme, but present a theoretical framework (figure 2) that, even though has some concepts related to SEL core elements (Durlak, Weissberg, Dymnicki, Taylor, & Schellinger, 2011; Jones & Bouffard, 2012) differs from it in a way that is not explained in the manuscript (i.e. why is empathy considered apart from socio emotional competence?). Authors should clarify how this program complies with SEL framework, in order to avoid confusion of terms and concepts in scientific research.

Thus, references are needed to support these affirmations in page 5: “The majority of SEL programme developers suggest that an instructionally sound and developmentally focused SEL curriculum should include elements of emotional education, cognitive restructuring, interpersonal problem-solving, social skills training, empathy training, problem-solving, stress reduction and relaxation, and behavioural change. It is believed that combining brief doses of each of these core elements in an instructionally sound manner would lead to improved SEL outcomes”

- Methodology of the programme: There is not enough information about the development of the programme itself: content of sessions, addressing of socio emotional competences, role of teachers, participants and dogs, picture books utilization…etc. Authors should give more details about the actual development of the programme, especially regarding the “canine-assisted reading component with self-developed picture books”, since this is presented as the main novelty that wants to be highlighted: When, where and how were the sessions carried out? What did the dogs and their trainers do? What were the children asked for? How were the picture books used? An example session should be presented in order to fully comprehend the characteristics of the programme.

It is also unclear what traditional “humane education” is, and how CARing kids differ from it, besides the animal component. The objective in the program as stated in the abstract is “to promote a human attitude and social-emotional competence”, so clarification about what is and what is not “a human attitude” is needed in order to make it possible for readers to understand why CARing kids is likely to provoke an effect on it. This applies also to control groups, especially in the “elite training group” which received various undefined educational strategies, remaining unclear their addressing of promotion of human attitude and socio emotional competence.

- Results and presentation of data: Authors are advised to be careful into claiming the positive effects of the CARing kids programme, since quantitative analysis were not so conclusive. Size effects should be calculated and presented for a better interpretation of the programme’s effect on the outcomes (Sullivan & Feinn, 2012). This could alter interpretations of the results, so this is a major concern. Also, authors should consider that the Elite Training control group showed significant results too, and include their interpretations about it.

Moreover, qualitative analysis methodology and results could be illustrated with a table or figure that includes the codes and list of themes collected by researches, to help readers understand the evaluation process and outcomes.

- Discussion: Authors seem to attribute the beneficial effects of the program to the animal presence, but this was not consistently evaluated through the process and is based only on their interpretations of qualitative analysis. Again, this issue could be related to the lack of information about the animal role in the programme, and what are authors referring to when they speak about “positive human-animal interaction”.

This lack of evaluation of the animal interaction factor is a limitation to the study that should be addressed by authors, including the statement in page 26 regarding number of dogs “The out-numbered human-to-dog ratio resulted in limited interaction time between participants and service animals.” (p.26). If it is so, it seems unlikely that human-animal interaction was the key to the positive effects observed. This applies to the underlying mechanisms among human-animal bonding that affect SEL competence too (p.31), because there is no measurable variable in this study that proves that socio emotional competence was affected mainly by the dog’s role and not by any other factors.

- Conclusion: The two first paragraphs are confusing, since authors here stress the impact of urbanization to justify the relevance of this study, but in the introduction section the development and application of the One Health concept through humane education is presented as the theoretical basis of the programme. Authors should clarify the purpose and justification of the study, correlating introduction and conclusion ideas, including a better exposition about what is humane education, why is it important and how is it being addressed within the CARing kids programme.

References

Durlak, J. A., Weissberg, R. P., Dymnicki, A. B., Taylor, R. D., & Schellinger, K. B. (2011). The impact of enhancing students’ social and emotional learning: A meta‐analysis of school‐based universal interventions. Child Development, 82(1), 405-432. doi:https://doi.org/10.1111/j.1467-8624.2010.01564.x

Jones, S. M., & Bouffard, S. M. (2012). Social and emotional learning in schools: From programs to strategies and commentaries. Social Policy Report, 26(4), 1-33.

Pérez-González, J. C., & Qualter, P. (2018). Emotional intelligence and emotional education in school years. In L. Dacree Pool, & P. Qualter (Eds.), An introduction to emotional intelligence (pp. 81-104). Chichester: Wiley.

Sullivan, G. M., & Feinn, R. (2012). Using effect size—or why the P value is not enough. Journal of Graduate Medical Education, 4(3), 279-282.

Zeidner, M., Roberts, R. D., & Matthews, G. (2002). Can emotional intelligence be schooled? A critical review. Educational Psychologist, 37(4), 215-231. doi:https://doi.org/10.1207/S15326985EP3704_2

Reviewer #2: The work is novel and rigorous, both at a theoretical and methodological level.

It is well structured, and represents an advance in the importance of developing socio-emotional skills ib children. The results are well described and are in accordance with the objectives and the methodological design.

6. PLOS authors have the option to publish the peer review history of their article (what does this mean?). If published, this will include your full peer review and any attached files.

Reviewer #1: No

Reviewer #2: **Yes: **Carmen Ferrándiz García

---

## [Author Response · Author response to Decision Letter 0]

21 Jan 2021

Dear Dr. Pérez-González, 

Thank you for giving us an opportunity to revise our manuscript. The comments and suggestions offered by you and the reviewers are very helpful and constructive. We have included the reviewer comments and responded to each comment individually. The changes were track-changed, and a cleaned manuscript was also submitted. Once again, we would like to thank you for your continued interest and valuable support in our research.

Regards, 

Paul W.C. WONG

Associate Professor and Registered Clinical Psychologist (HKPS) 

Department of Social Work and Social Administration, HKU 

Journal Requirements:

1. Please ensure that your manuscript meets PLoS ONE's style requirements, including those for file naming.

Response: We have double checked the PLoS ONE's style requirement and make sure that the manuscript is following PLoS ONE format

2. Please provide additional information about your teaching intervention, such as detailed curriculum, description of texts or methods used, or supporting educational material (such as the story books) that would allow others to replicate your study. If materials, methods, and protocols are well established, authors may cite articles where those protocols are described in detail, but the submission should include sufficient information to be understood independent of these references (https://journals.plos.org/plosone/s/submission-guidelines#loc-materials-and-methods). Please note that any materials submitted should not be under copyright more restrictive than CC-BY.

Response: We have included more information about the lesson plan. A sample session of the programme as support file.

3. We note that you have indicated that data from this study are available upon request. PLoS ONE only allows data to be available upon request if there are legal or ethical restrictions on sharing data publicly. For more information on unacceptable data access restrictions, please see http://journals.plos.org/plosone/s/data-availability#loc-unacceptable-data-access-restrictions.

Response: We have uploaded the dataset and deleted the identity information of the participants.

Response: We have moved the ethics statement and related sections to the methods session.

Reviewers' comments

5. Review Comments to the Author

Reviewer #1: This manuscript presents very interesting hypotheses about socio emotional learning strategies and their applicability in school-based programs, linked to a thorough research methodology. Authors have made an excellent use of quantitative and qualitative analysis to verify not only the effectiveness of the program, but its feasibility and sustainability over time, including context-related factors in the process. Furthermore, language, structure and presentation of data show the quality expected in a rigorous scientific journal.

Nevertheless, some aspects should be reviewed before publication:

- SEL theoretical framework: As SEL has proven to be effective in abundant research, it has also been criticized by its excessive broadness and variation between studies, which hinders validity and replication (Pérez-González & Qualter, 2018; Zeidner, Roberts, & Matthews, 2002). Authors state to choose SEL as the "theoretical foundation" of the CARing Kids programme, but present a theoretical framework (figure 2) that, even though has some concepts related to SEL core elements (Durlak, Weissberg, Dymnicki, Taylor, & Schellinger, 2011; Jones & Bouffard, 2012) differs from it in a way that is not explained in the manuscript (i.e. why is empathy considered apart from socio emotional competence?). Authors should clarify how this program complies with SEL framework, in order to avoid confusion of terms and concepts in scientific research.

Response: Many thanks for pointing out this issue. We have added a section of "Comparison between SEL, Humane Education and CARing Kids" to illustrate the difference between SEL, Humane education and CARing kids and explain this setting.

Thus, references are needed to support these affirmations in page 5: "The majority of SEL programme developers suggest that an instructionally sound and developmentally focused SEL curriculum should include elements of emotional education, cognitive restructuring, interpersonal problem-solving, social skills training, empathy training, problem-solving, stress reduction and relaxation, and behavioural change. It is believed that combining brief doses of each of these core elements in an instructionally sound manner would lead to improved SEL outcomes"

Response: We have further explained this statement and included extra-reference to support the statement.

- Methodology of the programme: There is not enough information about the development of the programme itself: content of sessions, addressing of socio emotional competences, role of teachers, participants and dogs, picture books utilization…etc. Authors should give more details about the actual development of the programme, especially regarding the "canine-assisted reading component with self-developed picture books", since this is presented as the main novelty that wants to be highlighted: When, where and how were the sessions carried out? What did the dogs and their trainers do? What were the children asked for? How were the picture books used? An example session should be presented in order to fully comprehend the characteristics of the programme.

Response: A section called "the overall structure of the CARing Kids" was added on page 8 to describe the curriculum further. Lesson 1 of the CARing Kids curriculum was attached in the Supplementary information to illustrate how the CARing kids were implemented. We also revised the result session to answers some implementation information 

It is also unclear what traditional "humane education" is, and how CARing kids differ from it, besides the animal component. The objective in the program as stated in the abstract is "to promote a human attitude and social-emotional competence", so clarification about what is and what is not "a human attitude" is needed in order to make it possible for readers to understand why CARing kids is likely to provoke an effect on it. This applies also to control groups, especially in the "elite training group" which received various undefined educational strategies, remaining unclear their addressing of promotion of human attitude and socio emotional competence.

Response: We have included the definition of humane attitude and included a session to compare the humane education and CARing Kids, and briefly describe why CARing Kids is likely to provoke the SEL and humane attitude. We combine this part with the newly written section of "Comparison between SEL, Humane Education and CARing Kids" on page 6 to 7. For the Elite training group, it is a routine school extra-curriculum activity. We have included more information of Elite training group in methods in Page 10. 

- Results and presentation of data: Authors are advised to be careful into claiming the positive effects of the CARing kids programme, since quantitative analysis were not so conclusive. Size effects should be calculated and presented for a better interpretation of the programme's effect on the outcomes (Sullivan & Feinn, 2012). This could alter interpretations of the results, so this is a major concern. Also, authors should consider that the Elite Training control group showed significant results too, and include their interpretations about it.

Response: Thank you for pointing out this issue. Cohen's d was calculated for each intervention conditions and embedded in Table 3. We also tune down about our writing on claiming the positive effect of CARing Kids programme. 

Moreover, qualitative analysis methodology and results could be illustrated with a table or figure that includes the codes and list of themes collected by researches, to help readers understand the evaluation process and outcomes.

Response: A new table 5 was included to show the codes, subthemes and themes to illustrate how the themes have emerged from the focus group data.

- Discussion: Authors seem to attribute the beneficial effects of the program to the animal presence, but this was not consistently evaluated through the process and is based only on their interpretations of qualitative analysis. Again, this issue could be related to the lack of information about the animal role in the programme, and what are authors referring to when they speak about "positive human-animal interaction".

This lack of evaluation of the animal interaction factor is a limitation to the study that should be addressed by authors, including the statement in page 26 regarding number of dogs "The out-numbered human-to-dog ratio resulted in limited interaction time between participants and service animals." (p.26). If it is so, it seems unlikely that human-animal interaction was the key to the positive effects observed. This applies to the underlying mechanisms among human-animal bonding that affect SEL competence too (p.31), because there is no measurable variable in this study that proves that socio emotional competence was affected mainly by the dog's role and not by any other factors.

Response: We have adjusted some part of the discussion, including the reviewer's reason and putting it into the limitation and further study. Although the student focus group result suggested that their intention to interact with readings dogs might contribute to their learning motivation, we also agree that more data were needed to support our statements. 

- Conclusion: The two first paragraphs are confusing, since authors here stress the impact of urbanization to justify the relevance of this study, but in the introduction section the development and application of the One Health concept through humane education is presented as the theoretical basis of the programme. Authors should clarify the purpose and justification of the study, correlating introduction and conclusion ideas, including a better exposition about what is humane education, why is it important and how is it being addressed within the CARing kids programme.

Response: Introduction was revised to link up the idea of urbanization, one health and the CARing Kids. The conclusion part was modified to increase the linkage and cohesiveness with the introduction. 

---

## [Decision Letter · Decision Letter 1]

10 Mar 2021

Effectiveness of a school-based programme of animal-assisted humane education in Hong Kong for the promotion of social and emotional learning: A quasi-experimental pilot study

PONE-D-20-25086R1

Dear Dr. Wong,

We’re pleased to inform you that your manuscript has been judged scientifically suitable for publication and will be formally accepted for publication once it meets all outstanding technical requirements.

Kind regards,

Gwo-Jen Hwang

Academic Editor

PLOS ONE

Reviewers' comments:

Reviewer's Responses to Questions

**Comments to the Author**

1. If the authors have adequately addressed your comments raised in a previous round of review and you feel that this manuscript is now acceptable for publication, you may indicate that here to bypass the “Comments to the Author” section, enter your conflict of interest statement in the “Confidential to Editor” section, and submit your "Accept" recommendation.

Reviewer #1: All comments have been addressed

2. Is the manuscript technically sound, and do the data support the conclusions?

Reviewer #1: Yes

3. Has the statistical analysis been performed appropriately and rigorously? 

Reviewer #1: Yes

4. Have the authors made all data underlying the findings in their manuscript fully available?

Reviewer #1: Yes

5. Is the manuscript presented in an intelligible fashion and written in standard English?

Reviewer #1: Yes

6. Review Comments to the Author

Reviewer #1: Authors have properly addressed all the comments raised in the first review. The article is now suitable for publication.

---

## [Editor Report · Acceptance letter]

12 Mar 2021

PONE-D-20-25086R1 

Effectiveness of a school-based programme of animal-assisted humane education in Hong Kong for the promotion of social and emotional learning: A quasi-experimental pilot study 

Dear Dr. Wong:

I'm pleased to inform you that your manuscript has been deemed suitable for publication in PLOS ONE. Congratulations! Your manuscript is now with our production department. 

Kind regards, 

on behalf of

Dr. Gwo-Jen Hwang 

Academic Editor

PLOS ONE